# Combined impact of pesticides and other environmental stressors on animal diversity in irrigation ponds

**Hiroshi C. Ito**[1,2]*, **Hiroaki Shiraishi**[1], **Megumi Nakagawa**[1], **Noriko Takamura**[1]

**1** National Institute for Environmental Studies, Tsukuba, Ibaraki, Japan, **2** Department of Evolutionary Studies of Biosystems, The Graduate University for Advanced Studies (Sokendai), Hayama, Kanagawa, Japan

* hiroshibeetle@gmail.com

**Data Availability Statement:** All relevant data are within the manuscript and its Supporting Information files.

## Abstract

Rice paddy irrigation ponds can sustain surprisingly high taxonomic richness and make significant contributions to regional biodiversity. We evaluated the impacts of pesticides and other environmental stressors (including eutrophication, decreased macrophyte coverage, physical habitat destruction, and invasive alien species) on the taxonomic richness of freshwater animals in 21 irrigation ponds in Japan. We sampled a wide range of freshwater animals (reptiles, amphibians, fishes, mollusks, crustaceans, insects, annelids, bryozoans, and sponges) and surveyed environmental variables related to pesticide contamination and other stressors listed above. Statistical analyses comprised contraction of highly correlated environmental variables, best-subset model selection, stepwise model selection, and permutation tests. Results showed that: (i) probenazole (fungicide) was a significant stressor on fish (i.e., contamination with this compound had a significantly negative correlation with fish taxonomic richness), (ii) the interaction of BPMC (insecticide; also known as fenobucarb) and bluegill (invasive alien fish) was a significant stressor on a "large insect" category (Coleoptera, Ephemeroptera, Hemiptera, Lepidoptera, Odonata, and Trichoptera), (iii) the interaction of BPMC and concrete bank protection was a significant stressor on an "invertebrate" category, (iv) the combined impacts of BPMC and the other stressors on the invertebrate and large insect categories resulted in an estimated mean loss of taxonomic richness by 15% and 77%, respectively, in comparison with a hypothetical pond with preferable conditions.

## Introduction

Freshwater ecosystems provide a broad variety of services, including disturbance regulation, water regulation, water supply, waste treatment, food production, and recreation [1], some of which are irreplaceable [2]. Although freshwater habitats contain only 0.01% of the world's water and cover only 0.8% of the Earth's surface [3], they maintain almost 6% of all described species and one-third of all vertebrate species [4, 5]. Among the various types of ecosystems,

**Funding:** N.T. acknowledges the support of the Environment Research and Technology Development Fund (S9) of the Ministry of the Environment, Japan. The funders had no role in study design, data collection and analysis, decision to publish, or preparation of the manuscript.

**Competing interests:** The authors have declared that no competing interests exist.

however, freshwater ecosystems have the highest proportion of species threatened with extinction [6, 7]. Because the loss of biodiversity tends to exponentially reduce the efficiencies and temporal stabilities of ecosystem functions [8], the current rapid biodiversity loss in freshwater ecosystems implies that they are degrading at a critical rate.

Major stressors on freshwater biodiversity include overexploitation, water pollution, flow modification, destruction or degradation of habitat, and invasion by alien species [4, 9]. Pesticide contamination is a major component of water pollution [10, 11]. Pesticides can have a serious impact on biodiversity due to their widespread application to reduce target animals, plants, and fungi in farmlands, which may affect non-target organisms as well. Experimental studies have shown that pesticide contamination decreases freshwater biodiversity [12]. While pesticide contamination in the environment is known to dramatically change community compositions into those dominated by pesticide-tolerant species [13–15], only recently has a significant negative relationship between pesticide concentrations and biodiversity been reported in freshwater invertebrates [16, 17]. Two issues make it difficult to evaluate pesticides' impacts on freshwater biodiversity in the environment as compared to experimental ecosystems. First, considering the spatiotemporal scale of pesticide application and residual effects, gathering reliable measurements of the states of communities and environmental variables at each sampling point is not easy, because many freshwater bodies have continuous inflows and outflows of organisms and water. Second, freshwater communities in the environment are affected by various environmental variables other than pesticides. Neglecting any of those non-pesticide variables can cause large uncertainties in the statistical evaluation of pesticides' impacts, if the neglected factor has a strong effect. Conversely, if we take into account all the environmental variables that have strong effects, we can reduce the uncertainties not only of pesticides' impacts but also the combined impacts of pesticides and other environmental stressors.

To overcome the first problem, we focused on irrigation ponds for rice cultivation, which are relatively closed and small systems in comparison with rivers and lakes and thus enable more reliable measurements of community states and environmental variables. Japan has approximately 200,000 irrigation ponds, most of which were constructed during the 17th to 19th centuries [18]. Despite their small size and the high risk of pesticide contamination and other stressors [19–22], the irrigation ponds can potentially sustain high taxonomic richness and make significant contributions to regional biodiversity [23–26]. Further, many endangered species inhabit the irrigation ponds [27]; the ponds function as refuges for various aquatic plants and wetland animals, because 61.1% of wetlands had already been lost by 2000 in Japan [18]. In this study, we sampled a wide range of freshwater vertebrates (reptiles, amphibians, and fish) and macroinvertebrates (mollusks, crustaceans, insects, annelids, and bryozoans) in 21 irrigation ponds of Hyogo Prefecture, Japan. Kadoya et al. [19] reported that biodiversity of the irrigation ponds in this region is at great risk of eutrophication, invasion of alien species, and physical habitat destruction, but the study did not investigate pesticide contamination.

To cope with the second problem described above, we surveyed 47 environmental variables corresponding to various stressors, including pesticide contamination, eutrophication, physical habitat destruction, decreased macrophyte coverage, and invasive alien species. We statistically analyzed the relationships between taxonomic richness of animals and environmental variables by means of model selection among multivariate regression models. Numerous explanatory variables (environmental variables), however, can cause not only a multicollinearity problem but also extremely heavy calculation for model selection procedures. To handle these difficulties, we developed a new statistical procedure by combining the contraction of explanatory variables (by using correlations among them), best-subset model selection,

stepwise model selection, and permutation tests. The developed procedure enabled us to detect previously unknown and significantly negative effects of two pesticides, probenazole (fungicide) and (2-butan-2-ylphenyl) N-methylcarbamate (BPMC [fenobucarb]; insecticide), on the taxonomic richness of the sampled animals and to evaluate the combined impacts of BPMC and other environmental stressors.

## Sampling and measurement

### Study area

Our study area covers approximately 580 km$^2$ in southwestern Hyogo Prefecture, Japan (34˚ 49′N, 134˚55′E). Predominant land uses are paddy fields, broad-leaved forests, and urban areas. The study area has a warm climate with a mean annual temperature of 14.4 ˚C (minimum 3.5 ˚C in January, maximum 26.4 ˚C in August) and mean annual precipitation of 1198.3 mm [19]. We selected 21 ponds to cover all typical land uses around the ponds, with surface areas ranging from 1935 to 22,163 m$^2$, depth ranging from 0.3 to 4.83 m, and elevation ranging from 10 to 130 m a.s.l. None of these 21 ponds had extraordinal overgrowth of macrophytes [28] during the study period.

### Sampling of vertebrates and macroinvertebrates

Sampling was conducted twice at each pond. At the first sampling (19 September to 5 October 2006), a fyke net (double 3-m wings, funnel 3.04 m, height 0.69 m, 4-mm nylon mesh) was set during daytime, with its two leaders set at the shore and the approximate center of the pond, respectively. Also, five rectangular bait traps (length 40 cm, height 25 cm, width 25 cm, 4-mm nylon mesh, mouths on both sides with 6-cm diameter, fish sausages and dried squid for bait) were set equally spaced along a line from shore to shore passing through the deepest point. The fyke net and traps were retrieved the following day. The second sampling (14–24 May 2007) was conducted near the shore with a D-frame dipnet (0.2-mm mesh) by 0.5-m-long discrete sweeps at 3 to 13 representative habitats (areas of floating-leaved plants, emergent plants, and leaf litter), depending on the pond's habitat diversity. Animals sampled with the fyke net and dipnet were identified to the lowest possible taxon. At this sampling, bottom surface sediment was collected three times at the approximate center of each pond with an Ekman–Birge-type sampler (mouth opening of 150 mm × 150 mm; Rigo, Tokyo, Japan). The collected sediment was washed through 0.2-mm mesh to eliminate the finer particles, and the samples were preserved in 10% formalin and identified to the lowest possible taxon under a binocular microscope. If an identified taxon included another identified taxon (e.g., one was a genus and another was species belonging to that genus), we assumed that they actually belonged to different lowest taxa from each other. In total, 144 taxa were identified (S1 Table).

The identified taxa included four invasive alien species: bluegill, *Lepomis macrochirus*; largemouth bass, *Micropterus salmoides*; red swamp crayfish, *Procambarus clarkii*; and bullfrog, *Lithobates catesbeianus*. These organisms are regulated under the country's Invasive Alien Species Act, meaning they are regarded to have the potential to harm ecosystems in Japan through predation on and competition with indigenous species (https://www.env.go.jp/en/nature/as.html). To evaluate their impacts as well as those of other stressors on freshwater animals in the studied ponds, these four invasive species were excluded and were instead treated as environmental variables that can influence biodiversity. We also excluded the pest insects *Galerucella nipponensis* and *Elophila interruptalis* collected on the agricultural crop water shield, *Brasenia schreberi* [29], since their responses to pesticides may be qualitatively different from those of other, non-pest animals.

For the remaining 138 taxa (hereafter, the "all-sampled" category), we counted the number of taxa in each pond. The all-sampled category was divided into seven subcategories: (1) reptiles, 3 taxa; (2) fishes, 13 taxa; (3) mollusks, 11 taxa; (4) crustaceans, 7 taxa; (5) large insects (Coleoptera, Ephemeroptera, Hemiptera, Lepidoptera, Odonata, Trichoptera), 48 taxa; (6) small insects (Diptera), 28 taxa; and (7) annelids (annelids, bryozoans, and sponges), 28 taxa. We separated the insects into two categories because the sampled dipterans consisted mainly of Chironomidae (23 of 28 taxa), a family that is known to be tolerant of water pollution [30], and thus may have a qualitatively different response to environmental variables than those of other insect orders. We referred to the last category simply as "annelids" because it consisted mainly of annelids (23 of 28 taxa).

## Environmental variables

For each pond, we measured 37 physicochemical water properties seven times in 2007 (April 23–24, May 28–29, June 18–19, July 17–18, August 13–14, September 3–4, September 25–26). The measured properties were water temperature, pH, total nitrogen, total phosphorus, suspended solids, chlorophyll *a*, and the concentrations of 31 pesticides (*insecticides*: BPMC, buprofezin, clothianidin, dinotefuran, fipronil, imidacloprid, malathion, tebufenozide, thiamethoxam; *fungicides*: azoxystrobin, ferimzone, fthalide, urametpyr, IBP, isoprothiolane, metominostrobin-E, metominostrobin-Z, probenazole, pyroquilon, thifluzamide, tiadinil, TPN; *herbicides*: bentazone, bromobutide, butachlor, chlomeprop, dymron, mefenacet, oxaziclomefon, pentoxazone, pyriminobac-methyl-E). See S1 Appendix 1 and S2 Table for details of the measurements, and see S3 and S4 Tables for the data. For each pesticide, concentrations lower than the detection limit were replaced with the detection limit concentration. All pesticides except for TPN were detected in at least one pond (S1, S2, and S3 Figs). In the statistical analysis, for each pond we used the maximum detected concentration among the seven samples for each of the 30 pesticides detected, and we used the average for each of the other six environmental variables. We also measured the organic matter content (ignition loss) in each pond's sediment once (13–15 May 2007) (S1 Appendix 2).

At each pond, we also measured the following 10 variables: pond depth, pond area, concrete bank rate (proportion of pond bank covered by concrete dike), percent coverage of floating-leaved plants, percent coverage of emergent plants, pond drainage intensity (0: no drainage, 1: partial drainage, 2: full drainage; see also [31]), and presence of the four invasive species: bluegill, largemouth bass, red swamp crayfish, and bullfrog (1: found, 0: not found). See S1 Appendix 2 for details of the measurements. The values of the environmental variables are summarized in S5 Table.

Among the environmental variables measured, the declines of floating-leaved plant coverage and that of emergent plant coverage may be stressors on the taxonomic richness of freshwater animals in the studied ponds. This is because macrophytes in irrigation ponds in the study area have been decreasing due to urbanization [32], an increase in concrete banks [33], and herbicide contamination. Some of the studied ponds had high concentrations of two herbicides, butachlor and pentoxazone (S2 Fig), which were far higher than their acute toxicity levels for the ecotoxicological bioindicator *Raphidocelis subcapitata* (72-h ErC50, 3.15 μg/L [34] and 0.79 μg/L [35], respectively). The decline of macrophytes can drive decadal change in benthic invertebrates [36]. To clarify this viewpoint, we transformed the percentages of floating-leaved plant coverage and emergent plant coverage into the area percentages not covered by these types of plants, as follows: 100 –(floating-leaved plant coverage) and 100 –(emergent plant coverage), respectively. Hereafter, we refer to these as "F-plant noncoverage" and "E-plant noncoverage", respectively.

Too shallow water depth causes unstable environments for freshwater animals, which may result in low biodiversity [37, 38]. Japanese irrigation ponds have been maintained through the periodic drainage and removal of bottom mud by farmers [31]. But recently the drainage and mud dredging have tended to be less frequent than in the past, and sometimes ponds are abandoned because of a decline in rice farming and farmers' aging [33]. These phenomena usually induce ponds to become shallower and eventually vanish [39]. Thus, we also transformed the depth of each pond into a shallowness index = (maximum depth among ponds)– (focal pond depth).

The numbers of observed taxa may have been affected by variation in the number of dipnet samples among ponds. However, normalization of the observed taxonomic richness by fitting rarefaction curves [16] was not appropriate for our data, because choices of sampling points and sampling numbers were both nonrandom; that is, they were designed to cover the existing habitat diversity with a minimum sampling number in each pond. As an alternative to normalization, we added the logarithm of dipnet sampling number to the 47 environmental variables, taking into account that sampling efforts and species numbers tend to show log–log relationships [40]. In total, 48 environmental variables were used in the statistical analysis (S5 Table).

## Ethics statement

We obtained permits for the survey from each pond manager in conjunction with the Agricultural and Environmental Affairs Department, Hyogo Prefecture Government. Surveyed ponds did not involve protected areas and species that required permits for sampling. The sampled invasive alien species were processed in accordance with the Japanese IAS Act. All native vertebrates were released into the same water bodies immediately after being measured and weighed.

## Statistical analysis

To identify which of the 48 environmental variables are related to the taxonomic richness (i.e., numbers of taxa) of the sampled animals, we conducted model selection among regression models and permutation tests. The response variables for the regression models were the taxonomic richness of the all-sampled category and seven subcategories (reptiles, fishes, mollusks, crustaceans, large insects, small insects, and annelids). In addition, to extract as much information as possible from our data, we classified taxa into four more categorizes, namely large animals (reptiles, fishes, mollusks, crustaceans, and large insects), small animals (small insects and annelids), vertebrates (reptiles and fishes), and invertebrates (mollusks, crustaceans, large insects, small insects, and annelids), and analyzed these as response variables as well. The analysis was conducted with statistical software R (version 3.4.4) and its packages glmmML-1.0, glmperm-1.0–5, spdep-0.7–9, pforeach-1.3, and foreach-1.4.4 (organized into R package "contselec," available from https://github.com/yorickuser/contselec).

## Contraction of environmental variables

The environmental variables were scaled so that their means and standard deviations became equal to 0 and 1, respectively. To reduce the amount of calculation needed and to avoid the multicollinearity problem, environmental variables with high absolute correlations were grouped together (by choosing 0.52 as the threshold for absolute value of correlation). This operation reduced the 48 environmental variables to 11 contraction groups. Nine of the groups contained a single variable: BPMC (insecticide), probenazole (fungicide), shallowness, F-plant noncoverage, concrete bank, pond drainage, bluegill, red swamp crayfish, and bullfrog; we refer to these as "single variables." The remaining two contraction groups were a small group

**Table 1. Contracted environmental variables.**

|  | Name | Description |
|---|---|---|
| Single variable | BPMC | concentration of insecticide BPMC |
|  | Probenazole | concentration of fungicide Probenazole |
|  | Shallowness | (maximum depth among ponds)–(focal pond depth) |
|  | F-plant noncoverage | 100 – (F-plant coverage rate) |
|  | Concrete bank | proportion of pond bank covered by concrete dike |
|  | Pond drainage | pond drainage intensity (0: no drainage, 1: partial drainage, 2: full drainage) |
|  | Bluegill | found(1)/unfound(0) of invasive alien fish *Lepomis macrochirus* |
|  | Red swamp crayfish | found(1)/unfound(0) of invasive alien crayfish *Procambarus clarkii* |
|  | Bullfrog | found(1)/unfound(0) of invasive alien frog *Lithobates catesbeianus* |
| Grouped variable | IBP-Ignition_loss | first principal component axis for the contraction group consisting of IBP (fungicide) and Ignition loss |
|  | Cont.var1, Cont.var2, Cont.var3, and Cont.var4 | first four principal component axes for the contraction group consisting of Largemouth bass (*Micropterus salmoides*), WT (water temperature), pH, SS (suspended solids), Chla (chlorophyll a), TP (total phosphorus), TN (total nitrogen), Area (pond area), E-plant noncoverage, D-net sampling, 8 insecticides (Buprofezin, Clothianidin, Dinotefuran, Fipronil, Imidacloprid, Malathion, Tebufenozide, and Thiamethoxam), 6 fungicides (Azoxystrobin, Ferimzone, Fthalide, Furametpyr, Isoprothiolane, Metominostrobin-E, Metominostrobin-Z, Pyroquilon, Thifluzamide, and Tiadinil), 9 herbicides (Bentazone, Bromobutide, Butachlor, Chlomeprop, Dymron, Mefenacet, Oxaziclomefon, Pentoxazone, and Pyriminobac-methyl-E) |

"IBP-Ignition_loss" consisting of IBP (fungicide) and ignition loss, and a large group containing the remaining 37 environmental variables. Each of these two groups was represented by its principal component analysis (PCA) axes so that more than 65% of its total variance was explained by the PCA scores (see also [41]). For the small group, only the first PCA axis was used (77.1% explained). For the large group, its top four PCA axes (65.5% explained) were used (S2 Appendix 1). We refer to these five representative variables as "grouped variables."

Consequently, the 48 uncontracted environmental variables were reduced to 14 contracted environmental variables, which included 9 single variables and 5 grouped variables (Table 1 and S6 Table). (In this analysis, we also tried integration of the pesticides' effects by calculating their toxic units [14, 16]. However, the integrated toxic units, $TU_{max}$ and $TU_{sum}$, both resulted in their belonging to the large contraction group, and then the effects of pesticides were not detected.).

Environmental variables with high absolute correlations were grouped together (by choosing 0.52 as the threshold for absolute value of correlation) in order to reduce the amount of calculation needed and to avoid the multicollinearity problem. Each group was represented by its principal component analysis (PCA) axes so that more than 65% of its total variance was explained by the PCA scores. Those PCA axes were called "grouped variables." In the statistical analysis, the single and grouped environmental variables were all scaled to range from 0 to 1. (See S6 Table for the data).

## Model selection

We used the 14 contracted environmental variables as the explanatory variables to explain the response variable, taxonomic richness of a focal animal category. For convenience, all explanatory variables were rescaled to range from 0 to 1 (their mean and standard deviation could differ from 0 and 1after this rescaling). For each of the possible subsets of the 14 explanatory variables, we constructed a Poisson regression mixed model [42], where any model has at least one explanatory variable. In each model, the response variables were described by a vector $\mathbf{y} = (y_1, \ldots, y_M)$ of length $M = 21$ (the number of studied ponds), where $y_i$ is its value for the $i$th pond. Explanatory variables were described by a set of vectors $\mathbf{x}_1, \ldots, \mathbf{x}_K$ with $1 \leq K \leq 14$, each

of which was denoted by $\mathbf{x}_k = (x_{k,1}, \ldots, x_{k,M})$. We assumed that $y_i$ follows the Poisson distribution,

$$y_i \sim \text{Poisson}(Y_i) \tag{1}$$

with its mean $Y_i$ described as

$$\ln(Y_i) = \alpha + \sum_{k=1}^{K} \beta_k x_{k,i} + r_i, \tag{2}$$

where $\alpha$ is the intercept, $x_{k,i}$ is the intensity of the $k$th explanatory variable at the $i$th pond with its regression coefficient $\beta_k$, and $r_i$ is a pond-specific random effect. $r_i$ follows the normal distribution with average 0 and standard deviation $\sigma$. For each of the models constructed above, we calculated maximum likelihood estimations for $\alpha, \beta_1, \ldots, \beta_K$, maximum marginal-likelihood estimation [42] for $\sigma$, and the Akaike information criterion (AIC = -2[maximum log-likelihood]+2[number of free parameters]) [43]. To suppress the estimation bias of AIC as a distance measure from an unknown true model, we excluded models that had more free parameters than one-third of the sample size [44]. We also fitted the normal Poisson regression model by setting $\sigma = 0$ in advance. We referred to the model with the lowest AIC as the contracted best model (see S2 Appendix 2 for details).

## Statistical inference

If the $p$-value for the regression coefficient of a focal explanatory variable is calculated by comparing the best model with its reduced model (generated by removing the focal variable from the best model) without taking into account the model selection conducted beforehand, then the calculated value is not an appropriate $p$-value for the null hypothesis that the focal explanatory variable has no effect on the response variable. This is because the model selection process affects the $p$-value for the null hypothesis [45]. In this study, we calculated the $p$-value corresponding to a null hypothesis that a focal explanatory variable has no negative effect (i.e., a one-sided test) by using a permutation test that specifically operates the model selection for each of 1000 resampled datasets (S2 Appendix 3). However, this permutation test requires extremely heavy calculation. Thus, to efficiently search for explanatory variables with statistically significant negative effects, we first looked for their candidates, referred to as statistically contributive explanatory variables, and then applied the permutation test to examine the significance of those candidates' effects. Specifically, we judged that a focal explanatory variable is statistically contributive when the variable satisfies the following three conditions: (i) The focal explanatory variable is included in all models of $\Delta\text{AIC} \leq C_{\Delta\text{AIC}}$ with $C_{\Delta\text{AIC}} = 2.0$ (i.e., differences in AIC from the contracted best model do not exceed 2.0), and its regression coefficients in those models have the same sign. (ii) In the contracted best model, the $p$-value for the regression coefficient of the focal explanatory variable is smaller than $\alpha_{\Delta\text{AIC}} = 0.05$ based on the permutation of regressor residuals test [46]. (iii) The focal explanatory variable is also included (keeping its sign) in the uncontracted best model that is chosen by the stepwise model selection by AIC among models composed of environmental variables before contraction, where the contracted best model is used as the initial model.

Among the three conditions above, condition (i) is the most important, and conditions (ii) and (iii) suppress biases due to small sample sizes and contraction of explanatory variables, respectively. In condition (i), the threshold $C_{\Delta\text{AIC}} = 2.0$ is chosen because any model with $\Delta\text{AIC} > 2.0$ is rejected by the parametric likelihood ratio test for significance level 0.05, when that model is nested in the contracted best model (see also [47]). Although this relationship does not hold for non-nested models, we consider choosing 2.0 to be a good strategy for

finding the candidates for explanatory variables with significant effects (see S2 Appendix 4 and S2 Appendix 5 for details). When a focal animal category had more than one statistically contributive explanatory variable in the above analysis (for main effects), we further analyzed interactions among them (see S2 Appendix 6 for details).

## Results

### Sampled animals

Fig 1 shows the number of taxa in each pond, ranging from 9 (21th pond) to 59 (1th pond) with mean 27.8 and SD 10.2 (see S7 Table for the data). The average frequency of each animal category was 4.8% for reptiles, 10.9% for fishes, 5.1% for crustaceans, 6.2% for mollusks, 16.1% for large insects, 33.4% for small insects, and 23.3% for annelids.

### Detected effects of environmental stressors on taxonomic richness

With regard to the taxonomic richness of the all-sampled category and its 11 subcategories, we found statistically significant effects of BPMC (insecticide), probenazole (fungicide), concrete bank, bluegill, F-plant noncoverage, and shallowness, all of which were negative. (see S8 Table for the calculated $p$-values, and S2 Appendix 7 for the best models). For convenience and brevity, we refer to the explanatory variables with statistically significant negative effects as "significant stressors." The relationships between those stressors and the taxonomic richness were intuitively represented in S4 Fig by biplot diagram of redundancy analysis (RDA).

Although BPMC and probenazole were not significant stressors on the all-sampled category (Fig 2a), BPMC was one of three significant stressors (BPMC, F-plant noncoverage, and

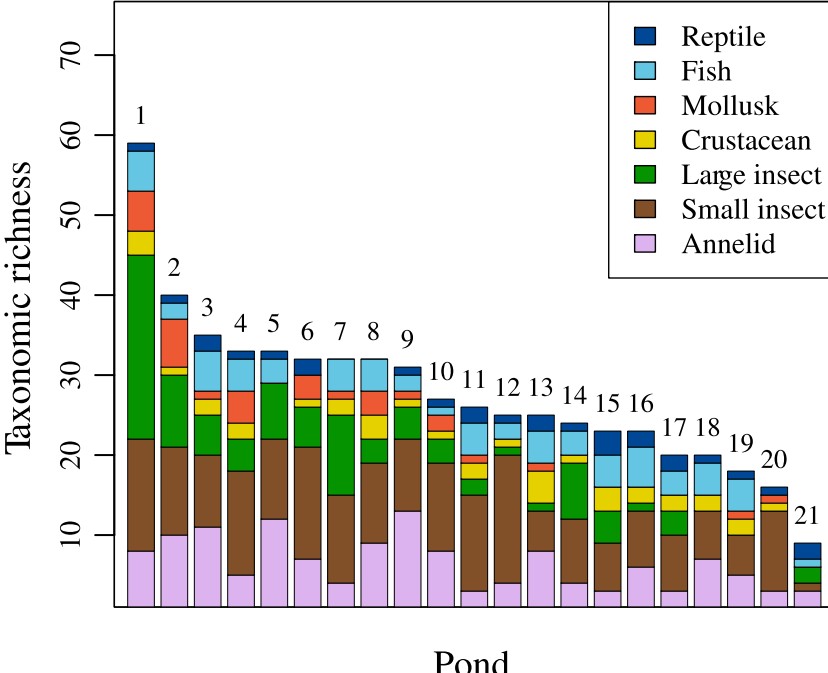

**Fig 1. Taxonomic richness of freshwater animals sampled in the study ponds.** The numbers atop bars are pond IDs assigned according to taxonomic richness. The large insect category consists of Coleoptera, Ephemeroptera, Hemiptera, Lepidoptera, Odonata, and Trichoptera. The small insect category consists of Diptera. The annelid category consists mainly of annelids and contains small fractions of bryozoans and sponges.

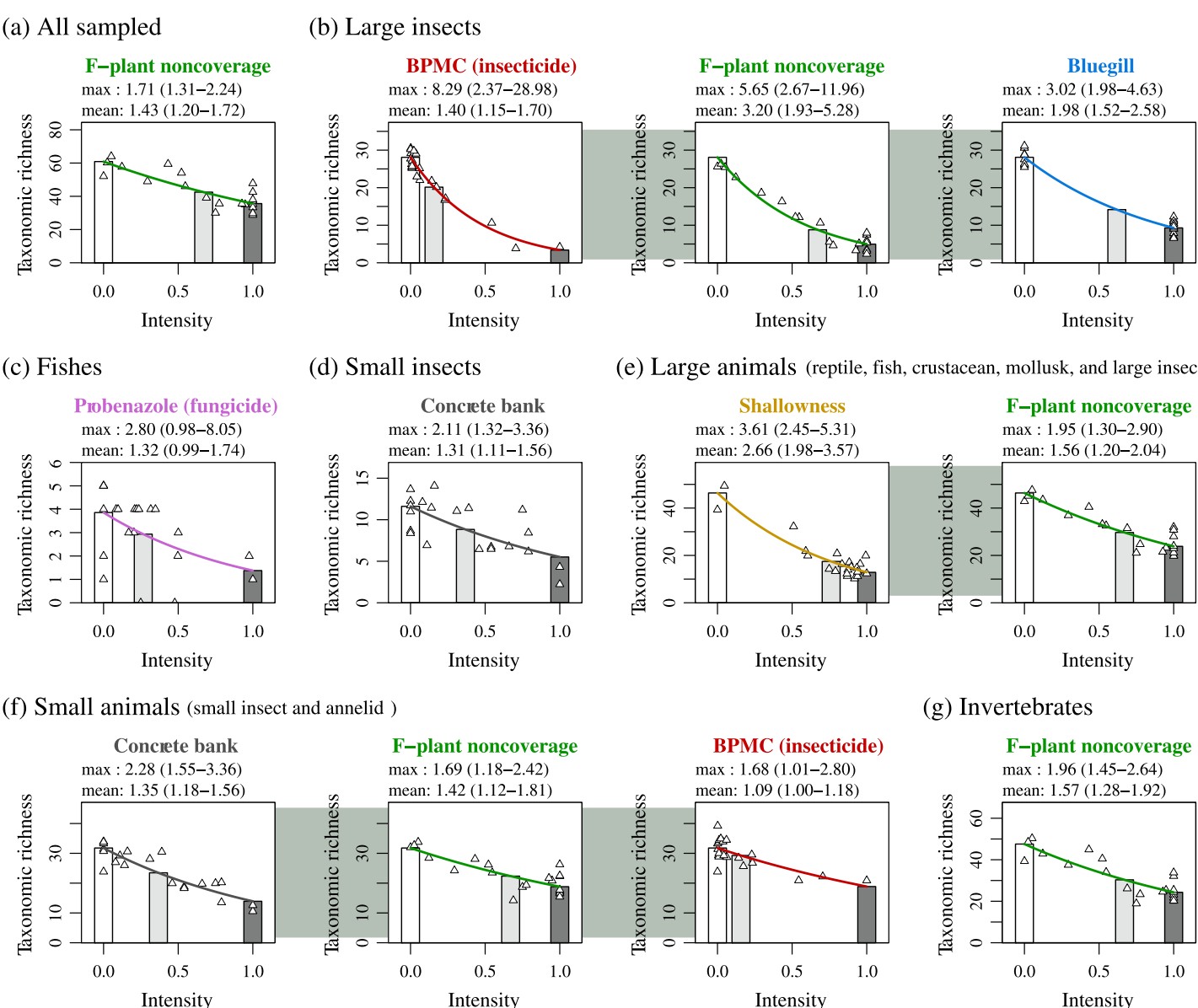

**Fig 2. Statistically significant stressors on taxonomic richness of all-sampled category and its subcategories: Large insects (Coleoptera, Ephemeroptera, Hemiptera, Lepidoptera, Odonata, and Trichoptera), fishes, small insects (Diptera), large animals (reptiles, fishes, mollusks, crustaceans, and large insects), small animals (small insects and annelids), and invertebrates (mollusks, crustaceans, large insects, small insects, and annelids).** In each panel, the white bar indicates the expected taxonomic richness of the focal animal category in the absence of all statistically contributive stressors ($R$ in Eq. (S2.10) in S2 Appendix 8). The light gray (or dark gray) bar indicates the expected taxonomic richness in the presence of only the focal stressor denoted by $x_j$ at its mean intensity $\bar{x}_j$ (or maximum intensity, scaled to 1.0) among the studied ponds, given by $R \exp(\beta_j \bar{x}_j)$ (or $R \exp(\beta_j)$) with its regression coefficient $\beta_j$ in the contracted best model. The value labeled with "mean" (or "max") shows the mean (or maximum) impact of the focal stressor among ponds, given by the height ratio of the white bar to the light gray bar (or dark gray bar). Specifically, the mean (or maximum) impact was calculated as $R/(R \exp(\beta_j \bar{x}_j)) = \exp(-\beta_j \bar{x}_j)$ (or $\exp(-\beta_j)$). (See S2 Appendix 8 for details). The estimation errors were calculated as Wald 95% confidence intervals, indicated in the format of (lower bound—upper bound). The solid curve indicates the expected taxonomic richness as a function $R \exp(\beta_j x_j)$ of the focal stressor's intensity $x_j$. The scatter plots indicate $R \exp(\beta_j x_{j,i}) + \varepsilon_i$, where $x_{j,i}$ is the intensity of the focal stressor at the $i$th pond, and $\varepsilon_i$ is the fitting residual of the contracted best model for the $i$th pond.

bluegill) on the large insect subcategory (Fig 2b), and probenazole was a significant stressor on the fish subcategory (Fig 2c). As for the other subcategories (reptiles, mollusks, crustaceans, small insects, and annelids), only small insects had a significant stressor, concrete bank (Fig 2d).

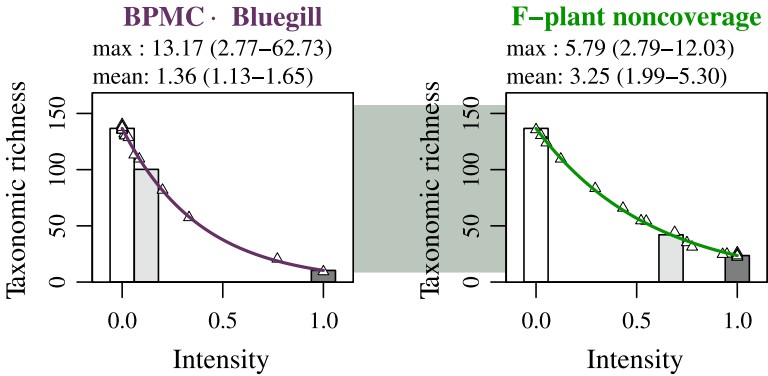

(a) Large insects

(b) Small animals

(c) Invertebrates

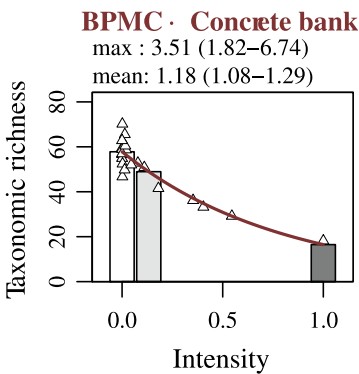

**Fig 3. Statistically significant interactions among stressors on taxonomic richness of categories of large insects (Coleoptera, Ephemeroptera, Hemiptera, Lepidoptera, Odonata, and Trichoptera), small animals (small insects (Diptera) and annelids), and invertebrates (mollusks, crustaceans, large insects, small insects, and annelids).** Result of analysis for detecting interactions among statistically contributive stressors (S2 Appendix 6) is shown. The plotting was done as in Fig 2.

When considering the large animal category, shallowness and F-plant noncoverage were significant stressors (Fig 2e), whereas the small animal category had a different set of significant stressors: concrete bank, F-plant noncoverage, and BPMC (Fig 2f). The invertebrate category had a single significant stressor: F-plant noncoverage (Fig 2g), whereas the vertebrate category had no significant stressors (see S2 Appendix 7). See S2 Appendix 9 for discussion on our statistical method.

Further analysis of interactions among statistically contributive stressors revealed significantly positive interactions between BPMC and bluegill for the large insect category (Fig 3a) and between BPMC and concrete bank for both the small animal (Fig 3b) and invertebrate categories (Fig 3c).

Each panel in Figs 2 and 3 lists the mean and maximum impacts of the focal stressor among the ponds, respectively (see caption of Fig 2). Although the mean and maximum impacts have large estimation errors, probenazole and BPMC tended to have weak mean impacts but strong maximum impacts.

Our analysis indicates that probenazole contamination has diminished the taxonomic richness of the fish category to 1/(mean impact) = 1/1.32 at the mean among ponds and to 1/(max. impact) = 1/2.80 at the worst pond (Fig 2c). In other words, the expected mean and maximum

losses of the fish taxonomic richness caused by probenazole are $100 \times (1–1/1.32) = 24\%$ and $100 \times (1–1/2.80) = 64\%$, respectively.

As for BPMC, the contracted best models with interactions (Fig 3) had no higher AICs than the corresponding contracted best models without interactions (Fig 2), as explained in S2 Appendix 6. Thus, Fig 3 may be more suitable for the estimation of BPMC's impacts. For the large insect category (Fig 3a), the interaction effect of BPMC and bluegill had a mean impact of 1.36 (26% loss) and maximum impact of 13.17 (92% loss). For the small animal category (Fig 3b), the interaction effect of BPMC and concrete bank had a mean impact of 1.19 (16% loss) and maximum impact of 3.69 (73% loss). For the invertebrate category (Fig 3c), the interaction effect of BPMC and concrete bank had a mean impact of 1.18 (15% loss) and maximum impact of 3.51 (72% loss).

## Combined impact of statistically significant stressors

Multiple significant stressors were detected for the large insect, large animal, and small animal categories (Figs 2 and 3). Since Poisson regression models were used for the fitting, the impacts of those stressors are multiplicative (explained in S2 Appendix 8). Thus, the combined impacts (defined by Eq. (S2.13)) can be plotted as additive effects on a logarithmic scale, as shown in Fig 4.

Clearly, the stressors' combined impacts are much stronger than the impact of each alone. Note that the stressors in Fig 4a and 4d (main effects only) are all included in Fig 4b and 4e (with interaction), respectively, where some of the main effects are replaced by their interactions. Since the contracted best models with interactions are all as good as the corresponding contracted best models without interactions, we here focus on those with interactions (Fig 4b and 4e) for the large insect and small animal categories.

Fig 4b indicates that the three significant stressors (BPMC, bluegill, and F-plant noncoverage) diminish the taxonomic richness of the large insect category to 1/(mean impact) = 1/4.43 at the mean among ponds and to 1/(max. impact) = 1/13.17 at the worst pond. In other words, the expected mean and maximum losses of the taxonomic richness of the ponds are $100 \times (1–1/4.43) = 78\%$ and $100 \times (1–1/13.17) = 92\%$, respectively, in comparison with the hypothetical normal pond free from all statistically contributive stressors (see S2 Appendix 8 for the definition). Likewise, Fig 4c indicates that the two significant stressors (shallowness and F-plant noncoverage) diminish taxonomic richness of the large animal category to 1/4.15 (76% loss) at the mean among ponds and to 1/6.96 (86% loss) at the worst pond. Fig 4e indicates that the three significant stressors (BPMC, concrete bank, and F-plant noncoverage) diminish taxonomic richness of the small animal category to 1/1.61 (38% loss) at the mean among ponds and to 1/4.22 (76% loss) at the worst pond.

## Discussion

### Impact of pesticides

Our study suggests that probenazole (fungicide) is a stressor on fish taxonomic richness in the studied ponds. Probenazole is a benzothiazole fungicide widely used in Asia for the control of rice blast fungus (*Magnaporthe grisea*) in paddy fields [48]. Its acute toxicity levels for the fish *Cyprinus carpio*, the crustacean *Daphnia magna*, and the aquatic plant *Raphidocelis subcapitata* [49] are all more than 1000-fold the maximum detected concentration of 0.73 µg/L measured in this study. As for the chronic effects of probenazole on fishes, we found no relevant experimental or field study. In general, however, fungicides can have diverse lethal and sublethal chronic effects on fishes and affect their physiology, development, and behavior [50]. In addition, some fungicides exhibit significant toxicity only when combined with other pesticides

(a) Large insects

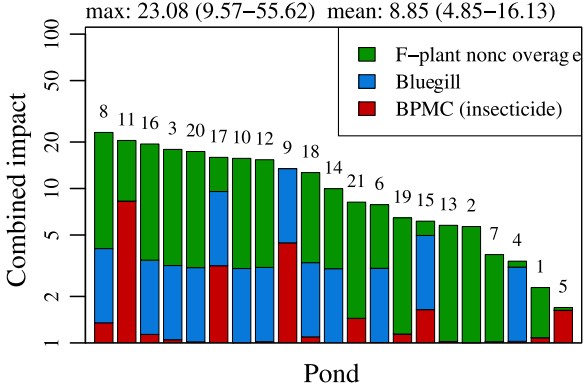

(b) Lartge insects (interaction)

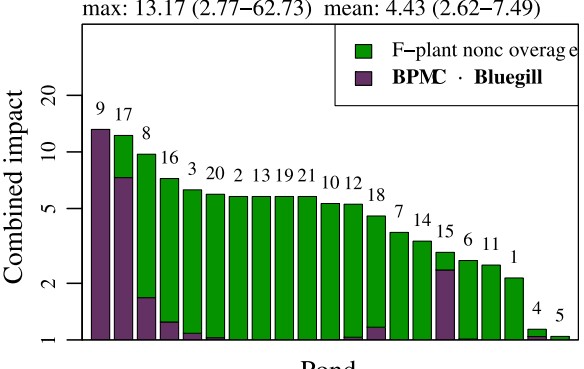

(c) Large animals

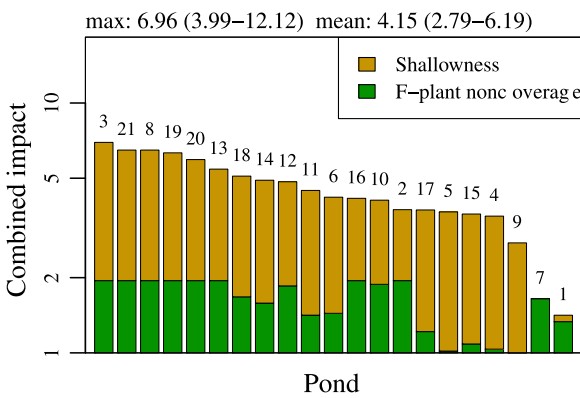

(d) Small animals

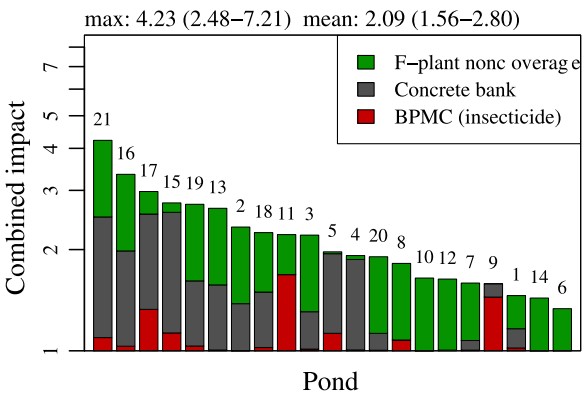

(e) Small animals (interaction)

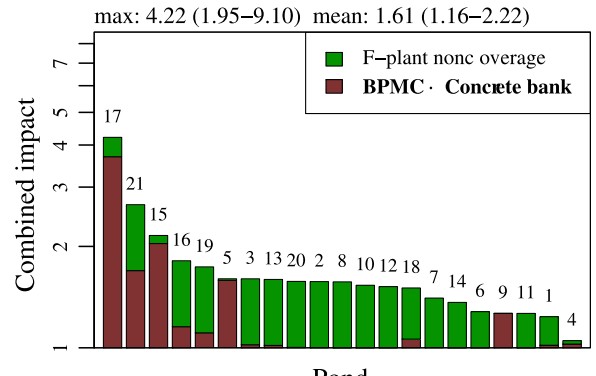

**Fig 4. Estimation of combined impacts of statistically significant stressors.** For each animal category that has multiple statistically significant stressors in Figs 2 and 3, the combined impact of those stressors in each pond is plotted as the reciprocal of the diminishing ratio of the taxonomic richness (S2 Appendix 8), by using the contracted best model (S2 Appendix 7). The numbers atop bars indicate pond IDs shown in Fig 1.

[51, 52]. Moreover, probenazole has a rapid decomposition rate (half-life of 9.8 h at pH 7 and 25 ˚C [49]) compared to our sampling frequency (once or twice per month), in which case the actual concentrations attained in the studied ponds could have been far higher than the detected concentrations. Therefore, our result may imply that probenazole actually has a negative impact on fish taxonomic richness. To clarify its impact, further experimental and field research is needed.

Our findings also suggest that BPMC is a stressor on the taxonomic richness of large insects (Coleoptera, Ephemeroptera, Hemiptera, Lepidoptera, Odonata, and Trichoptera), small animals (Diptera, annelids, bryozoans, and sponges), and invertebrates in the studied ponds. BPMC (fenobucarb) is a carbamate insecticide widely used in Asia to control rice planthoppers, but its impact on other invertebrates in the field is unclear. BPMC has a long half-life of 577 days (at pH 7 and 25 ˚C) [53], and its acute toxicity levels are 24-h EC50 = 10.2 µg/L for *D. magna*, 96-h LC50 = 25,200 µg/L for *C. carpio*, and 72-h EC50 = 33,000 µg/L for *R. subcapitata* (lowest values in [53]). However, even lower toxicity levels are reported for freshwater invertebrates: 96-h LC50 = 5.05 µg/L for the freshwater shrimp *Paratya improvisa* [54] and 48-h LC50 = 2 µg/L for the mayfly *Baetis thermicus* [55]. As for the chronic effect of BPMC, a concentration of 1 µg/L affects the development of the mayfly *Epeorus latifolium* [55]. Although 1 µg/L is still higher than the maximum concentration of 0.08 µg/L detected in our study, due to our once or twice monthly sampling the maximum concentration actually attained in the studied ponds could have been higher than 0.08 µg/L. Indeed, for pesticides in general, we can estimate from [16] (see Fig 2A) that the regional species richness of freshwater invertebrates would be reduced significantly when the detected pesticide concentrations attain 1/400th of their 48-h LC50 for *D. magna*. As for BPMC, its 48-h LC50 for *D. magna* is expected to be lower than its 24-h EC50 = 10.2 µg/L (because for *D. magna* the 48-h LC50 is essentially the same as the 48-h EC50, which must be lower than the 24-h EC50 = 10.2 µg/L). Thus, we can roughly estimate that invertebrate taxonomic richness in our studied ponds would decline at 10.2/400 = 0.026 µg/L of BPMC, which is less than the maximum detected concentration of 0.08 µg/L in our study. Therefore, our results for the large insect, small animal, and invertebrate categories accord with the results of [16] about pesticides' effects on regional invertebrate diversities.

Furthermore, BPMC contamination may also be affecting invertebrate taxonomic diversities in Japanese rivers, since far higher BPMC concentrations (5.6–37 µg/L [54, 56, 57]) have been reported from some of class A rivers (river systems directly administrated by Japanese government because of their importance for the national economy and people's lives). Yachi et al. [58] estimated the maximum BPMC concentrations (PEC$_{Tier2}$) at 350 river flow monitoring sites in 2010, using experimental data and the region-specific parameters of river flow, rice cultivation area, and pesticide usage ratio. From Fig 3 in [58], we can estimate that the upper 5% of those monitoring sites exceed 10 µg/L. Thus, invertebrates in Japanese rivers may be in a serious situation due to BPMC pollution.

In Japan, to prevent significant effects of a pesticide on aquatic organisms, pesticide registration standards are set based on acute toxicity test results of fishes, crustaceans, and algae. For pesticide registration (i.e., usage permission), the predicted environmental concentration (PEC) of the target pesticide must be lower than the registration standard of that pesticide [58]. Normally, PEC is calculated hierarchically according to the defined environmental model. For BPMC, its PEC of 2.1 µg/L according to the environmental model (PEC$_{Tier2}$ in [58]) was close to its registration standard of 1.9 µg/L, and PEC estimation from on-site monitoring data was permitted. Since the estimated value, 0.67 µg/L, was lower than the registration standard [53], the registration of BPMC has not been suspended, in other words, its application has not been restricted. This monitoring is expected to be conducted in accordance with

test guidelines for two sites where high concentrations are expected from pesticide use [53]. However, the maximum observed concentration of 0.67 μg/L is much lower than the 5.6– 37 μg/L reported for class A rivers [54, 56, 57]. Therefore, more monitoring sites in different regions may be needed to properly assess BPMC environmental concentrations in Japan, although use of BPMC in Japan has declined sharply since the 1990s, with the shipment volume of BPMC in 2015 representing only 3% of that in 1990.

With regard to the other 28 pesticides detected in our study, their relationships with taxonomic richness were unclear. In our statistical analysis, those pesticides had high correlations with other environmental variables (e.g., variables related to eutrophication), and thus they were contracted together and transformed into grouped variables. For statistical evaluation of those pesticides' impacts, we need to examine a different set of irrigation ponds than used in this study.

## Impacts of other statistically significant stressors

For the other statistically significant stressors detected in this study, previous studies support our results: see [19] for concrete bank, [59] and [19] for bluegill in irrigation ponds, [36] for lack of floating-leaved plant coverage in peatland drainage ditches, [37] for shallowness in floodplain lakes, and [38] for shallowness in ponds in an agricultural area. Among those studies, [19] surveyed irrigation ponds in the same region as our study, showing that not only concrete bank and bluegill but also chlorophyll *a* concentration was an important stressor on the taxonomic richness of freshwater animals. In our study, however, neither a statistically significant nor a contributive effect of chlorophyll *a* was detected. This difference may stem from the fact that our study considered pesticide contaminations and plant coverage as environmental variables, whereas [19] did not. In our study, the F-plant noncoverage was a statistically significant stressor, and it had a positive correlation ($r = 0.33$) with chlorophyll *a*, which may explain the difference at least in part.

Among the statistically significant stressors detected in our study, careful attention should be paid to the estimated impacts of shallowness and F-plant noncoverage. In this study, zero intensities for shallowness and F-plant noncoverage correspond to the maximum pond depth of 4.83 m and the highest F-plant coverage of 93%. In other words, we assumed that the all ponds originally had 4.83 m depth and 93% F-plant coverage, which may not necessarily correspond to their actual stress-free original states. However, their significant negative correlations with the taxonomic richness imply, at least, their potential as stressors, meaning that further increases in shallowness and F-plant noncoverage may decrease taxonomic richness. Conversely, if we can increase the water depth or F-plant coverages of those ponds, the taxonomic richness may recover.

## Combined impact of pesticides and other stressors

Our findings suggest that the taxonomic richness of freshwater animals in Japanese irrigation ponds has been affected by multiple significant stressors including pesticides. BPMC, F-plant noncoverage, and bluegill affect the large insect category (Figs 2b and 3a), shallowness and F-plant noncoverage affect the large animal category (Fig 2e), and BPMC, F-plant noncoverage, and concrete bank affect the small animal category (Figs 2f and 3b). According to [60], multiple stressors tend to act antagonistically, and therefore their cumulative mean effect is less than the sum of their single mean effects. In our analysis using the Poisson regression, when taxonomic richness was evaluated on a logarithmic scale (like the Shannon diversity index), a mean combined impact of multiple stressors was mathematically equal to the sum of their single mean impacts, as shown in Fig 4. On the other hand, when taxonomic richness was

evaluated on the normal scale, all of the mean combined impacts in Fig 4, except for the combined impacts on the large insect category, were weaker than the sum of the single mean impacts in Figs 2 and 3, in accordance with [60].

Our results show that the combined impact of BPMC and other significant stressors may have caused serious declines in taxonomic richness of the categories of large insect, small animal, and invertebrate, although our estimations have large uncertainties. We detected significantly positive interactions between BPMC and bluegill for the large insect category and between BPMC and concrete bank for the invertebrate and small animal categories. The former interaction is supported by an experimental study by Schulz and Dabrowski [61], who reported that the mortality of mayflies caused by insecticide exposure (azinphos-methyl and fenvalerate) synergistically increases with the presence of predatory fish. We found no relevant literature on the latter interaction.

The high sensitivity of the large insect category to the stressors may be partly due to its containing families of Ephemeroptera and Trichoptera, which, together with Plecoptera (not found in our sampling), are known to be highly sensitive to environmental degradation of freshwater systems [62–64]. In our study, the large insect category had positive correlation with the total taxonomic richness: $r = 0.85$ (large insect included) or $r = 0.55$ (large insect excluded), showing its potential as an indicator for animal biodiversity as well as environmental degradation of pond ecosystems.

Our survey of freshwater animals and environmental variables in irrigation ponds concluded that serious decline of taxonomic diversity in macroinvertebrates may have been caused by the combined impact of an insecticide (BPMC) and other stressors, including concrete bank protection, alien fish (bluegill) invasion, and lack of floating-leaved plants. Our results in conjunction with other literature imply that BPMC pollution has been seriously affecting invertebrates in Japanese freshwater systems.

## Supporting information

**S1 Code.**
(ZIP)

**S1 Appendix. Measurement of physicochemical properties of pond water.**
(PDF)

**S2 Appendix. Statistical analysis.**
(PDF)

**S1 Fig. Changes of insecticide concentrations in studied ponds.** In each panel, red, blue, and green indicate the top 3 ponds with the highest detected concentrations among the 21 ponds. The others are colored gray. Each point connecting line segments indicates one of the seven samplings during the study period.
(TIF)

**S2 Fig. Changes of herbicide concentrations in studied ponds.** The plotting was done as in S1 Fig.
(TIF)

**S3 Fig. Changes of fungicide concentrations in studied ponds.** The plotting was done as in S1 Fig. Among the 13 fungicides measured, TPN is not shown because it was not detected in any pond.
(TIF)

**S4 Fig. Biplot diagram calculated by redundancy analysis (RDA) on taxonomic richness of animal categories and detected stressors (environmental variables with significantly negative effects) in studied ponds.** Plotted numbers indicate pond IDs. 55.6% of total variance was explained by all RDA axis. The calculation was conducted by R (version 3.4.4) and its package "vegan" (version 2.5–3).
(TIF)

**S1 Table. Sampled animals.**
(XLSX)

**S2 Table. Chosen methods for pesticide extraction and measurement.**
(XLSX)

**S3 Table. Measured environmental variables (water qualities).**
(XLSX)

**S4 Table. Measured pesticide concentrations.**
(XLSX)

**S5 Table. Uncontracted environmental variables in studied ponds.** Mean and maximum values are based on seven measurements taken during the study period.
(XLSX)

**S6 Table. Contracted environmental variables in studied ponds.**
(XLSX)

**S7 Table. Taxonomic richness (number of taxa) of sampled animals in studied ponds.**
(XLSX)

**S8 Table. Calculated _p_-values (statistical significance) for effects of statistically contributive explanatory variables.** See S2 Appendix 3 for the algorithm and S2 Appendix 7 for the best models.
(XLSX)

## Acknowledgments

We thank the editors and two anonymous reviewers for valuable comments on earlier versions of this manuscript. We thank Y. Oikawa and A. Saji (National Institute for Environmental Studies, Tsukuba; NIES) for pretreatment of samples for water quality analysis; Y. Oikawa (NIES) for pretreatment of samples for pesticide analysis; S. Serizawa and I. Hirai (NIES) for assistance with pesticide analysis; T. Murakami (Regional Ecosystem Conservation), Y. Daihu (Regional Ecosystem Conservation), I. Murakami (Regional Environmental Planning Inc.), R. Ueno (NIES), A. Ohtaka (Nirosaki University), and U. Nishikawa (NIES) for animal sampling; M. Akasaka for investigation of peripheral land use and pond vegetation; and M. Imada (NIES) for interviewing pond owners about drainage.

## Author Contributions

**Conceptualization:** Noriko Takamura.

**Data curation:** Megumi Nakagawa.

**Formal analysis:** Hiroshi C. Ito, Hiroaki Shiraishi, Megumi Nakagawa.

**Funding acquisition:** Noriko Takamura.

**Investigation:** Megumi Nakagawa, Noriko Takamura.

**Methodology:** Hiroshi C. Ito, Hiroaki Shiraishi, Megumi Nakagawa, Noriko Takamura.

**Project administration:** Noriko Takamura.

**Software:** Hiroshi C. Ito.

**Supervision:** Noriko Takamura.

**Visualization:** Hiroshi C. Ito.

**Writing – original draft:** Hiroshi C. Ito.

**Writing – review & editing:** Hiroaki Shiraishi, Noriko Takamura.

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
