## [Decision Letter · Decision Letter 0]

16 Apr 2020

PONE-D-20-01257

Combined impact of pesticides and other environmental stressors on taxonomic richness of freshwater animals in irrigation ponds

PLOS ONE

Dear Dr. Ito,

Thank you for submitting your manuscript to PLOS ONE. After careful consideration, we feel that it has merit but does not fully meet PLOS ONE’s publication criteria as it currently stands. Therefore, we invite you to submit a revised version of the manuscript that addresses the points raised during the review process.

We would appreciate receiving your revised manuscript by May 31 2020 11:59PM. To enhance the reproducibility of your results, we recommend that if applicable you deposit your laboratory protocols in protocols.io, where a protocol can be assigned its own identifier (DOI) such that it can be cited independently in the future. For instructions see: http://journals.plos.org/plosone/s/submission-guidelines#loc-laboratory-protocols

We look forward to receiving your revised manuscript.

Kind regards,

Michael A Chadwick, PhD

Academic Editor

PLOS ONE

Additional Editor Comments (if provided):

Thank you for this interesting paper. The 2 reviewers were both positive about your work, but did suggest several areas which need your attention. The main areas which both identified were related to clarity in your Methods and Results. Both have given a clear list of suggested improvements which will need your attentions. I agree with their comments and strongly suggest attention be paid to your statistical methods and clarity in presenting your findings (both within the text and figures) in submitted revision.

Journal Requirements:

Reviewers' comments:

Reviewer's Responses to Questions

**Comments to the Author**

1. Is the manuscript technically sound, and do the data support the conclusions?

Reviewer #1: Yes

Reviewer #2: Partly

2. Has the statistical analysis been performed appropriately and rigorously? 

Reviewer #1: Yes

Reviewer #2: I Don't Know

3. Have the authors made all data underlying the findings in their manuscript fully available?

Reviewer #1: Yes

Reviewer #2: Yes

4. Is the manuscript presented in an intelligible fashion and written in standard English?

Reviewer #1: Yes

Reviewer #2: Yes

5. Review Comments to the Author

Reviewer #1: Biological monitoring in relation with pesticides and other chemicals is an important field not only from the viewpoints of science, but also for national task force on the resource and environmental management.

However, it is difficult since biological communities are receiving various different environmental factors simultaneously, as authors described well in the introduction of the present paper.

From this view point, the present paper provides meaningful statistical approaches on the inference of monitored data set.

In my opinion, the authors focused too much on the statistical inference regarding the explanatory variables, and missed some important general viewpoint for the responses of biological communities of ponds, although the category-level responses are described.

So, I just suggest some minor revisions as below,

1. Description of pond sites

Fig. 1 shows the taxonomic richness of monitored ponds.

It would be more understandable if authors explain the monitoring results regarding the the ponds having the highest richness (9) and the lowest one (17) briefly in the head of results section.

Their situations can tell the relationship between richness and environments directly as well as the importance of variables qualitatively.

2. The expected taxonomic richness of the normal pond

For the biological monitoring particularly for the management and restoration,

reference site (or standard) is very important for the estimation (same for the present results, figures 2, 3, and 4).

Is there any reference or available data that can show the reliability of this theoretical prediction?

Since it is very hard to find the sites having no stressors, this kind of information can be very helpful for the biological monitoring and its inference.

3. The response of biological communities.

Since the sites are located within Hyogo Prefecture (basic environments are similar), I think they basically have similar and overlapped species composition particularly for fish and some macroinvertebrates.

From the data obtained from the present research, I think the authors are able to extract some tendencies of species responses, regarding sensitive and insensitive species (for example density response or presence/absence response).

I believe that researchers in biological monitoring fields will be interested in the present results,

so, they will be interested in not only how to simplify the complicated explanatory variables but also in how to simplify the response variables such as using the necessary target species and indicators.

If the authors can suggest these kind of information briefly (if possible), the paper can provide more valuable information regarding the biological monitoring and set up the management strategy.

4. Other minor revision and questions

Line 135

for Micropterus salmodies, "largemouth bass" is better. As far as I know, black bass is often used for all species in genus Micropterus.

Line 235, 257

How the variables were scaled to 0 mean and range from 0 to 1?

Line 252

"such that the effects of pesticides were not clarified"

Does it mean "not clarified as real variable and not considered in the present analyses"?

Line 528, 544

What is the definition of class A rivers?

Reviewer #2: This is an interesting paper quantifying the effects of biotic and abiotic stressors on animal diversity in freshwaters. However, some of the statistical methods used seem a little strange to me, and I feel that the results have been overcomplicated as a result. I think the paper could be improved by simplifying the Methods and Results to focus on the main findings. My major concerns are:

1. The methods are very long, I think they could be simplified and some details moved to supporting information. I am not familiar with some of the statistical methods, and references are not given for these – please provide them. There are a lot of equations that are not well described, are they all necessary in the main text?

2. The stressors you are using are not clear from the outset, and the way they are referred to is often confusing and not clearly defined (e.g., “concrete bank”, “F-plant noncoverage”). Perhaps include a table to show the stressors and how they are grouped for your analyses? Try and use more informative names for the stressors in the text, with the details in the new table.

3. I am concerned about how the animals have been grouped for the analyses? Can this be justified? Additionally, many of the responses overlap (i.e. insects are also invertebrates). Why? Do you need all of these responses?

4. In the Results you refer to “statistically contributive negative effects” and

“statistically significant negative effects”. This is a bit odd – does it mean that the “statistically contributive negative effects” are not significant? If so, remove from the results and only concentrate on the significant effects for brevity. You show loss of richness and link it to some stressors, but where the richness is loss (i.e. what species) is not described. Would an ordination help?

Other comments:

Title: How about using “Multiple stressors impacts on animal diversity in irrigation ponds” instead? Or something else more succinct.

Line 20: What others stressors? Be more specific

Line 24-26: Remove details on statistics from Abstract, instead incorporate it into the sentences om results (see next point).

Line 27: what do you mean by “unique significant stressor”. The only stressor which had an effect? The results section of the Abstract is unclear. Be more specific and don’t overcomplicate. E.g. “The fungicide probenazole had a significant negative effect on fish species richness, while the insecticide BPMC interacted with fish invasion to cause a significant decline in the taxonomic richness of large insects”.

Line 31: First mention of ‘concrete bank’! What stressors are you including in your study?

Line 56, 60, 64: don’t say “in the field”

Line 86-93: Move to methods

Line 113: Is it possible that this wide variation in surface area and/or elevation could be driving the results you describe? Was this considered?

Line 124: Why different sampling procedures for the different points? It makes it hard to make comparisons.

Line 146: Across both sampling dates?

Line 155-157: Move to results

Line 165: I think this should come above the section on animal diversity sampling

Line 178: Why the maximum concentration and not the mean?

Line 184: But above you said they were no macrophytes?

Line 189-209: A lot of this should come in the Introduction

Line 217: That is a lot of environmental variables! Are they listed somewhere in the main text, not just SI?

Line 224-227: These groupings seem a little odd. Can you justify them?

Line 233: I am not familiar with this method to “contract” the predictor variables. Please justify it and give at least one reference.

Line 240: ‘real variables’ is difficult, are the others not real then?! How about ‘single variables’ and ‘grouped variables’ or something similar

Line 257: Repetitive, you have already said they are scaled

Line 272: Define all these terms (AIC etc). Also, I believe you should be using AICc (for small sample sizes)?

Line 298-306: References for these conditions?

Results: These are not very clear and have been over complicated. Use concise statements. For instance, the first sentence could be: “Two pesticides (Probenazole and BPMC), the presence of concrete banks, bluegill invasion, shallowness, and lack of macrophytes all had a negative effect on animal richness”

Line 491: Do these species occur in your sites?

Line 504: So these response variables overlap? i.e. insects are also invertebrates. It is necessary to have all of these overlapping responses? It overcomplicates what could be a very nice concise paper.

Line 602-623: I think this, and a lot of the details and equations on the statistics, should be moved to a supporting document. The main text can then be simplified and those interested in the details can refer to the SI. Remove this, and end with a concise conclusion instead.

Figure 1: Are the ponds ordered in some way here? I.e. by impacted to non-impacted? If not, this figure seems a bit pointless.

6. PLOS authors have the option to publish the peer review history of their article (what does this mean?). If published, this will include your full peer review and any attached files.

Reviewer #1: No

Reviewer #2: No

---

## [Author Response · Author response to Decision Letter 0]

25 May 2020

Dear Sir,

Thank you very much for reviewing our manuscript “Combined impact of pesticides and other environmental stressors on taxonomic richness of freshwater animals in irrigation ponds” (Ms. no. PONE-D-20-01257). We were happy to receive the encouraging feedback by the expert editor and reviewers. We have revised our manuscript in accordance with the editor’s and reviewers’ valuable comments and suggestions, which is specifically described in the submitted MS Word file "Cover_letter_"

2020_05_24.docx"

Sincerely,

Hiroshi C. Ito, Hiroaki Shiraishi, Megumi Nakagawa, and Noriko Takamura

---

## [Editor Report · Decision Letter 1]

16 Jun 2020

Combined impact of pesticides and other environmental stressors on animal diversity in irrigation ponds

PONE-D-20-01257R1

Dear Dr. Ito,

We’re pleased to inform you that your manuscript has been judged scientifically suitable for publication and will be formally accepted for publication once it meets all outstanding technical requirements.

Kind regards,

Michael A Chadwick, PhD

Academic Editor

PLOS ONE

Additional Editor Comments (optional):

Thank you for this improved submission. Having carefully evaluated the work it is clear that you have addressed all of the concerns raised by the initial review. I feel that this work is now acceptable for publication. However, I would suggest that the paper could be improved and be of interest to a wide audience if one minor addition was made. Specifically, I think an evaluation of your "a new statistical procedure by combining the contraction of explanatory variables" (line 90), included as a section in the Discussion, would be a great value. However as neither reviewer asked for this in the original submission, I feel that this is a suggestion to improve the paper rather than a requirement.

In addition, I would suggest moving the "Ethics Statement" to the end of the "Materials and methods" section, so readers can focus on your approach. Finally, the paper should be checked to ensure the proper formatting is used prior to publication.
---

## [Editor Report · Acceptance letter]

23 Jun 2020

PONE-D-20-01257R1 

Combined impact of pesticides and other environmental stressors on animal diversity in irrigation ponds 

Dear Dr. Ito:

I'm pleased to inform you that your manuscript has been deemed suitable for publication in PLOS ONE. Congratulations! Your manuscript is now with our production department. 

Kind regards, 

on behalf of

Dr. Michael A Chadwick 

Academic Editor

PLOS ONE